# A Symmetry-Aware Learning Approach for Solving Mixed-Integer Linear Programs

## Abstract

Recently, machine learning techniques have been widely utilized for solving mixed-integer linear programs (MILPs). Notably, learning-based approaches that encode MILPs as bipartite graphs and then leverage graph neural networks (GNNs) to identify high-quality solutions have shown remarkable potential. Symmetry as an intrinsic property of many MILPs leads to multiple equivalent solutions, which incurs noticeable computational challenges and hence is treated with care in classic optimization algorithms. However, to the best of our knowledge, none of the learning-based methods take special care of symmetry within MILPs such that their computational performance might be jeopardized. To mitigate this issue, we propose a symmetry-aware learning approach that includes (i) position embeddings as node features to differentiate interchangeable variable nodes, and (ii) a novel loss function to alleviate ambiguity caused by equivalent solutions. We conduct extensive experiments on public datasets and the computational results demonstrate that our proposed approach significantly outperforms existing ones in both computational efficiency and solution quality.

## 1 Introduction

Mixed integer linear programs (MILPs) are optimization problems that combine discrete and continuous variables, and have a wide range of practical uses across various fields, such as production planning (Pochet & Wolsey, 2006; Chen, 2010), resource allocation (Liu & Fan, 2018; Watson & Woodruff, 2011), and transportation management (Luathep et al., 2011; Schöbel, 2001). Variables in such problems can often be permuted without changing the structure of the problem, which is known as symmetry (Margot, 2003). Typical examples of problems with symmetry include bin packing (Kaibel & Pfetsch, 2008), graph coloring (Ramani et al., 2006), and job scheduling (Martínez et al., 2019).

The presence of symmetries in MILPs leads to the emergence of numerous equivalent sub-problems. This results in conventional optimization algorithms like branch and bound becoming inefficient, as they consume significant computational resources by exploring equivalent solutions.(Liberti, 2012). Some classic approaches were proposed to handle symmetries by fixing variables (Bendotti et al., 2021), adding symmetry-breaking inequalities (Margot, 2009), and pruning enumeration trees (Margot, 2002). Due to their efficiency and robustness (Pfetsch & Rehn, 2019), such techniques have been widely incorporated as a fundamental component in modern optimization solvers.

Graph neural networks (GNNs), owing to their expressive capabilities and generalization performance, have recently been applied to assist MILP-solving algorithms, and demonstrate their great potential for improvements (Nair et al., 2020). Specifically, a large class of approaches use GNNs to learn a mapping from a graph representation $\mathcal{G}$ of an MILP instance to its solution in a supervised learning fashion, see Figure 1: The input of GNN are the MILP instances encoded as bipartite graph $\mathcal{G}$, and the labels are the solutions obtained from commercial solvers. A GNN is trained given the above input and labels, and is used for solution prediction. The GNN prediction may not be feasible, and one should always search around the neighborhood of the predicted one to identify a high-quality solution (Nair et al., 2020; Han et al., 2023; Khalil et al., 2022). However, without considering symmetry properties, such methods have two major drawbacks when facing problems with significant symmetries:

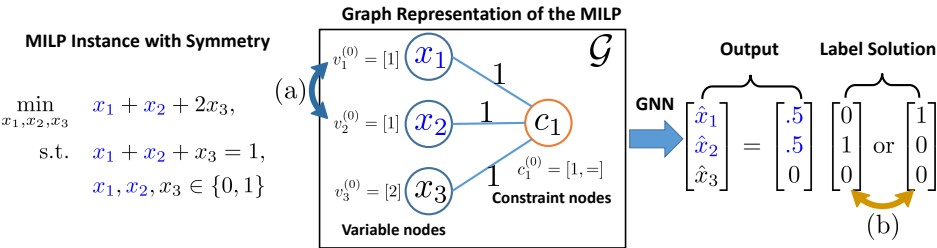

Figure 1: This figure shows a toy example of the GNN-based approaches and their drawbacks when solving MILPs with symmetry: (a) Indistinguishable variable nodes, (b) Label ambiguity.

(a) *Indistinguishable variable nodes.* Given two variable nodes with identical feature vectors, GNNs will give **same outputs for these indistinguishable nodes**. For instance, in Figure 1 (a), nodes $x_1$ and $x_2$ have the same node features and connectivity, such that GNNs cannot distinguish them and will produce the same output, $\hat{x}_1 = \hat{x}_2$. However, the desired prediction values may be different, $\hat{x}_1 \neq \hat{x}_2$.

(b) *Label ambiguity.* As shown in Figure 1 (b), an MILP instance with symmetry has multiple equivalent solutions, resulting in multiple labeling options. However, there is no general guideline on how to choose one of them as the label. Picking a suitable label is important, as it significantly affects the targets that the GNN fits, thereby impacting the final performance.

Consequently, existing approaches often exhibit modest or even poor performance when applied to problems with significant symmetries.

In this paper, we investigate a general symmetry type that allows permutations over groups of variables. Focusing on this symmetry, we propose a feature design of the graph node with position embeddings to differentiate indistinguishable variables and a new loss to alleviate the label ambiguity drawback. More specifically, for drawback (a), we encode positional information into the feature vector of each variable node. As a result, the model can output diverse values for those indistinguishable variable nodes. For the ambiguity drawback, we design a symmetry-aware loss and propose an algorithm that iteratively permutes training labels by minimizing their distances to the current prediction. To verify the effectiveness of the proposed method, we conduct extensive experiments on four public datasets. Our computational results demonstrate the proposed method can significantly outperform existing methods.

The distinct contributions of our work can be summarized as follows.

1. We propose to incorporate position embedding into the graph representation design to resolve drawback (a) that GNN can only output the same values for indistinguishable nodes.

2. We design a symmetry-aware loss function associated with an iterative algorithm that mitigates the ambiguity caused by multiple equivalent solutions.

3. We conduct comprehensive computational studies on four typical public datasets involving symmetries, and the results show that our proposed approach significantly outperforms existing methods.

## 2 RELATED WORKS

Previous works on identifying high-quality solutions to MILPs via machine learning techniques mainly focus on reducing problem sizes. Ding et al. (2020) propose to identify and predict a subset of decision variables that stay unchanged within the collected solutions. Li & Wu (2022) formulate MILPs as Markov decision processes and learn to reduce problem sizes via early-fixing.

It is noteworthy that the emergence of GNNs has had a significant impact on mixed-integer linear programming. Gasse et al. (2019) are the first to propose a bipartite-graph representation of MILPs and pass it to GNNs. Nair et al. (2020) adopt the same representation scheme and train GNNs to predict the conditional distribution of solutions, from which they further sample solutions. Rather than directly fixing variables, Han et al. (2023) conduct search algorithms in a neighborhood centered around an

initial point generated from the predicted distribution. Other works based on GNNs (Sonnerat et al., 2022; Lin et al., 2019; Khalil et al., 2022; Wu et al., 2021) also illustrate great potential in improving the solving efficiency.

Limitations of the existing GNN-based approaches are also noticed. Nair et al. (2020); Han et al. (2023) try to tackle the label ambiguity drawback by learning conditional distribution. Chen et al. (2022) introduce random features into the bipartite-graph representation to differentiate variable nodes but did not show the generalizability.

However, none of the existing learning-based approaches take special care of symmetry handling. On the contrary, works from mathematical perspectives suggest symmetry-handling algorithms exhibit great abilities in solving symmetry-involving MILPs (Pfetsch & Rehn, 2019) To name a few, such algorithms include: orbital fixing (Ostrowski et al., 2011), tree pruning (Margot, 2002), and lexicographical ordering (Kaibel & Pfetsch, 2008).

## 3 BACKGROUND

In this section, we define the problem we are interested in and describe two crucial components of the GNN-based approach proposed by Han et al. (2023), including the GNN structure and the neighborhood search algorithm.

### 3.1 PROBLEM STATEMENT

As the integer decision variables can be represented as a sequence of binary variables (Nair et al., 2020), without loss of generality, we will consider the following simplified mixed-binary linear programs:

$$
\begin{aligned}
\min_{x,y} \quad & w^{x\top}x + w^{y\top}y, \\
\text{s.t.} \quad & A\begin{bmatrix} x \\ y \end{bmatrix} \leq b, \quad y \in \mathbb{R}^{n_y}, \\
& x_{ij} \in \{0,1\}, \quad \forall i = 1, \ldots, s, j = 1, \ldots, q,
\end{aligned}
\tag{1}
$$

where the constraint coefficient $A \in \mathbb{R}^{m \times n}, b \in \mathbb{R}^m$, objective coefficients $w^x \in \mathbb{R}^{n_x}, w^y \in \mathbb{R}^{n_y}$, the decision variables $x \in \{0,1\}^{n_x}, y \in \mathbb{R}^{n_y}$, and $n = n_x + n_y = s \cdot q + n_y$. We assume problem (1) has the symmetries mentioned in chapter 17.7 of (Jünger et al., 2009): if $\pi$ is any permutation of the ground set $I^q := \{1, 2, \ldots, q\}$, then there exists a permutation $\pi'$ of $I^{n_y}$ such that

$$
\left( x_{11}, \ldots, x_{s1}, x_{12}, \ldots, x_{s2}, \ldots, x_{1q}, \ldots, x_{sq}, y_1, \ldots, y_{n_y} \right)
\tag{2}
$$

is feasible if and only if

$$
(\underbrace{x_{1\pi(1)}, \ldots, x_{s\pi(1)}, x_{1\pi(2)}, \ldots, x_{s\pi(2)}, \ldots, x_{1\pi(q)}, \ldots, x_{s\pi(q)}}_{\pi(x)}, y_{\pi'(1)}, \ldots, y_{\pi'(n_y)})
\tag{3}
$$

is feasible and both solutions have the same objective value. We denote $X$ as the matrix form of $x$, and use these two forms in the remaining parts according to the context. The symmetry described above simply means the columns of $X$ can be arbitrarily permuted.

### 3.2 BIPARTITE GRAPH REPRESENTATION FOR MILP

Gasse et al. (2019) proposed to represent a MILP by a bipartite graph $\mathcal{G} = (V, C, E)$ with two disjoint sets of nodes, $V = \{v_1, v_2, \ldots, v_n\}$ and $C = \{c_1, c_2, \ldots, c_m\}$, denoting the decision variables and constraints in Problem (1), respectively. And $E = \{A_{jk} | A_{jk} \neq 0, c_j \in C, v_k \in V\}$ is the set of weighted edges connecting variable nodes and constraint nodes, where $A$ is the coefficient matrix in Problem (1). Each node has a feature vector $v_k$ or $c_j$ describing the information of variables or constraints, e.g., variable types (continuous or binary), lower and upper bounds, right-hand-side coefficients, constraint types $(=, \leq, \geq)$, etc. See Figure 1 for the illustration and more details in Section 4.1.

### 3.3 Graph convolutional neural network

In the graph convolutional neural network (GCNN)-based approach proposed by Gasse et al. (2019), a bipartite graph $\mathcal{G} = (V, C, E)$ (with node features $c_j^{(0)} = c_j$, $v_k^{(0)} = v_k$, and edge features $A_{jk}$) is taken as the input. Stacked layers are applied to aggregate information from neighbors and update node embeddings. Each layer has two consecutive *half convolutions* computed as

$$c_j^{(l)} = f_c^{(l)} \left( c_j^{(l-1)}, \sum_{j:(j,k)\in E} g_c^{(l)} \left( c_j^{(l-1)}, v_k^{(l-1)}, A_{jk} \right) \right), \tag{4}$$

$$v_k^{(l)} = f_v^{(l)} \left( v_k^{(l-1)}, \sum_{k:(j,k)\in E} g_v^{(l)} \left( c_j^{(l)}, v_k^{(l-1)}, A_{jk} \right) \right), \tag{5}$$

where $l \in \{0, \cdots, L\}$ denotes the layer index, $f_c^{(l)}$ and $g_c^{(l)}$ are non-linear transformations gathering information from variable nodes and update on constraint nodes, $f_v^{(l)}$ and $g_v^{(l)}$ are on the contrary. All of these four transformations are two-layer perceptrons with ReLU activations. Lastly, another two-layer perceptron $f_{out}$ with sigmoid activation is used to convert the final embeddings to the predictions of integer variables by $\hat{x}_k = \text{Sigmoid}(f_{out}(v_k^{(L)}))$. We denote $f_\theta$ as the GNN parameterized by $\theta$, and the output vector for the discrete part as $\hat{x} = f_\theta(\mathcal{G})$ in the remaining sections.

### 3.4 Neighborhood search

Recent works (Ding et al., 2020; Han et al., 2023) incorporate neighborhood search algorithms with model predictions to identify high-quality solutions. In this work, we adopt the predict-and-search framework presented in Han et al. (2023). We first acquire the predicted solution $\hat{x}^{(i)}$ via the GNN model $f_\theta$. Then we record the indices of the smallest $k_0$ elements and the largest $k_1$ elements in sets $I_0$ and $I_1$, respectively, which are considered confidently predicted. Based on this, we define the $(k_0, k_1, \Delta)$-neighborhood :

$$\sum_{i \in I_0} x_i + \sum_{j \in I_1} 1 - x_j \leq \Delta, \tag{6}$$

where $x$ is the binary decision vector, and $\Delta$ restricts the radius of the neighborhood. It means that, for those confidently predicted variables, we do not expect to see too much deviation (restricted by $\Delta$). Adding (6) to the original problem, one can often obtain a better-quality solution through the solver.

## 4 Proposed method

In this section, we first describe the proposed bipartite graph feature design and symmetry-aware loss to mitigate drawback (a) and (b), respectively. Then, we propose the optimization scheme for training the GNNs model.

### 4.1 Position embeddings as node features

Let us first describe the node features generally used in the literature, and then introduce position embeddings used for differentiating indistinguishable variable nodes.

**General node features:** As described in Section 3.2, we encode a MILP instance into a bipartite graph $\mathcal{G}^{(i)}$. Each variable node $v_k$ accommodates a feature vector representing variable type (continuous or discrete), bounds, cost coefficients, and node degree. Meanwhile the $j$-th constraint node $c_j$ is encoded to include the constraint type (i.e. $=, \leq, \geq$), right-hand-side, and node degree. Edge features are designed to express the corresponding coefficient $A_{jk}$ between the $j$-th constraint and the $k$-th variable.

**Position embeddings:** To differentiate indistinguishable variables, we attach extra features to each variable node. In contrast to previous works (Chen et al., 2022), which add random noise to each node, we design the extra features according to the symmetry property of the MILP considered. Specifically, the assigned feature has no necessity to be different for every decision variable, which may cause severe generalization issue. Regarding the symmetry mentioned in Section 3.1, the feature should only be different for elements in different columns, but same for elements in the same row. For an MILP defined in (1) with all possible column-permutations, we construct the extra features of the $k$-th column as a $D$-dimensional vector in the form proposed by Vaswani et al. (2017):

$$\mathrm{PE}_k(d) := \begin{cases} \sin(k/10000^{2d/D}), & \text{if } d \text{ is odd,} \\ \cos(k/10000^{2d/D}), & \text{otherwise.} \end{cases}$$

where $\mathrm{PE}_k$ is the position embedding vector for column $k$, and $d$ denotes $d$-th element of the vector. For variables in $X$, $X_{ij}$ will get position embedding $PE_j$ as extra features. For variables that do not belong to $X$, we simply use a $D$-dimensional zero vector. By concatenating the general features and position features together, we get the final features of the bipartite graph, which is subsequently taken as the input of the GNN model.

## 4.2 SYMMETRY-AWARE LOSS

In our task, the most critical question for MILP is how to assign values to discrete decision variables, and we only target the prediction of binary variables, which are labeled by $x^{(i)} \in \{0, 1\}^{sq}$. The training dataset $\mathcal{D} = \{\mathcal{G}^{(i)}, x^{(i)}\}_{i=1}^N$ contains $N$ instances, where $\mathcal{G}^{(i)}$ is the bipartite graph representation of the $i$-th MILP. We use the exact GNN architecture $f_\theta(\cdot)$ as described in Section 3.3 to predict the corresponding output $\hat{x}^{(i)} = f_\theta(\mathcal{G}^{(i)})$.

We now define the **symmetry-aware loss** as follows:

$$\ell^\pi(\hat{x}^{(i)}, x^{(i)}) = \min_{\pi^{(i)}} \quad \ell\left(\hat{x}^{(i)}, \pi^{(i)}(x^{(i)})\right), \tag{7}$$

where $\ell(\cdot, \cdot)$ can be any distance-measuring function, e.g., the mean square error (MSE) or the binary cross entropy (BCE). This loss measures a minimum "distance" from all the possible permuted label solutions to the predictions. We then train the GNN predictor using the subsequent loss,

$$\mathcal{L}(\theta) = \frac{1}{N} \sum_{i=1}^N \ell^\pi(\hat{x}^{(i)}, x^{(i)}), \tag{8}$$

in which we have to optimize the GNN parameters $\theta$ and the label permutations $\{\pi^{(i)}\}_{i=1}^N$. This loss function provides a general guideline on the label selection.

## 4.3 MODEL TRAINING

In this section, we propose an iterative algorithm, that iteratively optimizes the GNN parameters $\theta$ and the label permutation $\{\pi^{(i)}\}_{i=1}^N$, see Algorithm 1. In each iteration, the GNN parameters $\theta$ are updated via gradient-based (GD) optimizers, and the update of $\pi^{(i)}$ can be solved in polynomial time by reformulating as a linear assignment problem. This algorithm can be interpreted as a dynamic relabeling: in each iteration, optimally perturbed labels $\hat{\pi}^{(i)}(x^{(i)})$ are used for the training.

---

**Algorithm 1** GNN training (full batch)

---

1: **Input:** Dataset $\mathcal{D} = \{\mathcal{G}^{(i)}, x^{(i)}\}_{i=1}^{N}$, learning rate $\alpha$, **Output:** $\theta_t$
2: Initialize $\theta_1$
3: **for** $t \leftarrow 1$ **to** $T$ **do**
4:    **for** $i \leftarrow 1$ **to** $N$ **do**
5:       $\hat{x}^{(i)} \leftarrow f_{\theta_t}(\mathcal{G}^{(i)})$
6:       $\hat{\pi}^{(i)} \leftarrow \arg\min_{\pi^{(i)}} \ell\left(\hat{x}^{(i)}, \pi^{(i)}(x^{(i)})\right)$
7:    **end for**
8:    $\mathcal{L}(\theta_t) \leftarrow \frac{1}{N} \sum_{i=1}^{N} \ell\left(\hat{x}^{(i)}, \hat{\pi}^{(i)}(x^{(i)})\right)$
9:    $\theta_{t+1} \leftarrow \mathrm{GD}(\theta_t, \nabla_{\theta_t} \mathcal{L}(\theta_t); \alpha)$
10:    **until** some stopping criteria are met
11: **end for**

---

**The update of** $\pi^{(i)}$    In the training loop, before computing the gradient, one has first to solve a batch of discrete optimization sub-problems (7), see line 6 in Algorithm 1. Note that these sub-problems can be solved in parallel. For ease of illustration, let us denote the permutation $\pi^{(i)}$ using its matrix-form $P^{(i)} \in \mathcal{P}$ instead, where $\mathcal{P}$ is a set of all the permutation matrices. The sub-problem (7) is rewritten as follows:

$$\min_{P^{(i)}} \quad \ell(\hat{X}^{(i)}, X^{(i)}P^{(i)}),$$
$$\text{s.t.} \quad P^{(i)}\mathbf{1} = \mathbf{1}, P^{(i)\top}\mathbf{1} = \mathbf{1}, \tag{9a}$$
$$P^{(i)} \in \{0,1\}^{q \times q}. \tag{9b}$$

When the loss function $\ell(\cdot, \cdot)$ is MSE or BCE,

$$\ell_{MSE}(\hat{X}^{(i)}, X^{(i)}P^{(i)}) = \|\hat{X}^{(i)} - X^{(i)}P^{(i)}\|_F^2 \tag{10}$$
$$= \mathrm{tr}(\hat{X}^{(i)\top}\hat{X}^{(i)} - 2\hat{X}^{(i)\top}X^{(i)}P^{(i)} + P^{(i)\top}X^{(i)\top}X^{(i)}P^{(i)}) \tag{11}$$
$$= \mathrm{tr}(\hat{X}^{(i)\top}\hat{X}^{(i)} - 2\hat{X}^{(i)\top}X^{(i)}P^{(i)} + X^{(i)\top}X^{(i)}), \tag{12}$$
$$\ell_{BCE}(\hat{X}^{(i)}, X^{(i)}P^{(i)}) = -\sum_{jk}([X^{(i)}P^{(i)}]_{jk} \log \hat{X}_{jk}^{(i)} + (1 - [X^{(i)}P^{(i)}]_{jk}) \log(1 - \hat{X}_{jk}^{(i)})), \tag{13}$$

the objective function is linear w.r.t. $P^{(i)}$. Then this problem is known as the *linear assignment problem* (Burkard & Cela, 1999), which can be efficiently solved to optimal by Hungarian algorithm (Kuhn, 1955) in polynomial time.

We finally note that Algorithm 1 can be easily adapted to a mini-batch version, where the data used in the inner loop can be a random mini-batch sampled from $\mathcal{D}$.

## 5   EXPERIMENTS

In this section, we present numerical studies on publicly available datasets along with detailed settings. The source code used in our experiments can be found in https://anonymous.4open.science/r/SA-MILP-CFFE/.

### 5.1   SETTINGS

**Benchmarks**   We evaluate our approach on four publicly available datasets, in which three involve symmetry mentioned in Section 1, and one has no such symmetry. The first dataset is the *Balanced Item Placement* (IP) from the NeurIPS ML4CO 2021 competition (Gasse et al., 2022), and we extract 400 problem instances for our experiments. The problem structure is fixed in each instance with 1050 binary variables and $q = 10$. The size of the training, validation, and testing sets are 240, 60,

and 100, respectively. The second dataset is the *Steel Mill Slab problem* (SMSP) from Schaus et al. (2011), and it contains 380 problem instances. The number of binary variables in the SMSP ranges from $22,422$ to $24,420$. We use $243$, $61$, and $76$ instances for training, validation, and testing sets, respectively. The third dataset is generated by adding random perturbation to instance assign1-10-4 in MIPLIB 2017 (Gleixner et al., 2021) (denoted by AP). In addition, we take *Workload Appointment problem* (WA) from (Gasse et al., 2022) as comparison, which has no symmetry aforementioned.

**Solution collection**    The above benchmarks only include problem instances. We collected the corresponding solutions from a commercial solver Gurobi Optimization, LLC (2023). For each instance, we run single-thread Gurobi for 3,600 seconds, and record the best solution along with their objective values. These solutions are further used as labels for training the baseline and our proposed approach, and the best-collected solution for the $i$-th instance is denoted as $\{X^{(i)}\}$.

**Baselines**    In the experiments, we use three methods as baselines, they are as follows. **Gurobi:** We use the commercial solver Gurobi to solve the benchmark instances under a single-thread setting. **PS:** Another baseline is the prediction-search framework from Han et al. (2023). Our proposed approach mainly differs from this framework in two aspects: **(i)** PS does not include positional information when encoding MILPs into bipartite graphs; **(ii)** PS learns a conditional distribution from multiple solutions, which is not symmetry-aware. **LEX:** We also include the traditional optimization approach, lexicographic ordering, as one of the baselines. This approach sorts columns of $X^{(i)}$ lexicographically in a non-increasing order (Friedman, 2007), and obtains the new label $\bar{x}^{(i)}$ such that $\bar{x}^{(i)} \succeq \pi(x^{(i)}), \forall \pi$. A more detailed description can be found in Section A of the supplementary materials.

**Training**    Our model is trained on a training set using Algorithm 1 with a mini-batch approach, running for 100 epochs. We use Adam optimizer with a learning rate of 0.001 for parameter learning, and other optimizer parameters are set to their defaults. We picked the model with the smallest symmetry-aware loss on the validation set for the evaluation in the next step.

**Evaluation**    We feed the GNN output of each algorithm to a neighborhood search module mentioned in Section 3.4 to evaluate the prediction quality. The neighborhood search process operates on a single thread for a maximum of 1000 seconds, as limited improvements of solution quality are observed after this time limit. The evaluation metric used in the experiments is the *primal gap*, i.e., $\mathrm{PG}(\tilde{x}, \tilde{y}) = |w^{x\top}\tilde{x} + w^{y\top}\tilde{y} - w^{x\top}x^* - w^{y\top}y^*|/(|w^{x\top}x^* + w^{y\top}y^*| + \epsilon)$, which measures the relative gap in the objective value of a feasible solution $(\tilde{x}, \tilde{y})$ to that of the best-known solution $(x^*, y^*)$, and $\epsilon$ is a small positive value to avoid the numerical issue. We refer to the best solution obtained from all the experimented approaches as the best-known solution (BKS). An average primal gap (APG) is calculated across all the test instances. The neighborhood search module has three hyper-parameters $(k_0, k_1, \Delta)$ that need to be tuned. To ensure a fair comparison among different algorithms, we employed a grid search strategy to find the best hyper-parameters for each one. Subsequently, these hyper-parameters were used for neighborhood search on various datasets to evaluate the prediction quality of different algorithms. The details of the grid search tuning are described in Section B of the supplementary materials.

**Computational resources and software**    All evaluations are performed under the same configuration. The evaluation machine has one AMD EPYC 7H12 64-Core Processor @ 2.60GHz, 256GB RAM, and one NVIDIA GeForce RTX 3080. Gurobi 9.5.2 and PyTorch 2.0.1 (Paszke et al., 2019) are utilized in our experiments. The time limit for running each experiment is set to $1,000$ seconds since a tail-off of solution qualities was often observed after that.

## 5.2 RESULTS

**Main results**    We first compare our proposed symmetry-aware approach with the baselines, using the APG ($\downarrow$) across test instances on three benchmarks with symmetry (IP, SMSP, AP), respectively, see Figure 2. Our proposed method significantly outperforms the baselines over the primal gap at 1,000 seconds. To be more specific, in the IP dataset, Gurobi and LEX exhibit comparable performance, being surpassed by both PS and our method. Initially, PS outperforms our method before 600 seconds. However, PS experiences a gradual performance over time, then is eventually surpassed by Ours

with a significant gap. For dataset AP and SMSP, our method dramatically outperform other three algorithms.

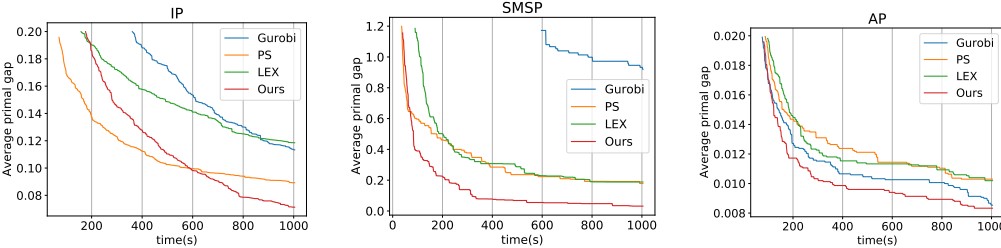

Figure 2: Performance comparisons between Gurobi, PS, LEX, and Ours, where the y-axis is the relative primal gap averaged across test instances; each plot represents one benchmark dataset.

We also report the average objective values given by different approaches at $1,000$ seconds in Table 1. We use $\text{gap}_{\text{abs}}$ to evaluate the performance of each approach, which is the absolute gap between the corresponding objective value and the objective value of BKS. The performance gain of ours is calculated by computing the relative gaps between our approach's absolute gaps and the smallest absolute gaps across Gurobi, PS and LEX.

In Table 1, we report the objective values obtained by different algorithms, and the extent of symmetry is also reported (denoted by $\Phi$), which indicates the number of all possible equivalent solutions. We remark that our method achieved $4\%$, $40\%$ and $69\%$ performance gain on AP, IP and SMSP datasets, respectively. It is noticeable that, though PS can produce legitimate solutions, our method achieves nearly equivalent performance compared to the BKS. Besides, Our method exhibits computational consistency by persistently outperforming other approaches on all datasets with symmetry. The overall results demonstrate the effectiveness of our method, which utilizes the position embedding and the symmetry-aware loss.

Table 1: Average objective values given by different approaches at 1,000 seconds.

| dataset | Obj. of BKS | Gurobi | | PS | | LEX | | Ours | | $\Phi$ | gain |
|---|---|---|---|---|---|---|---|---|---|---|---|
| | | obj. | $\text{gap}_{\text{abs}}$ | obj. | $\text{gap}_{\text{abs}}$ | obj. | $\text{gap}_{\text{abs}}$ | obj. | $\text{gap}_{\text{abs}}$ | | |
| WA | 699.1 | 699.2 | 0.10 | 699.1 | 0 | - | - | 699.1 | 0 | 0 | 0% |
| AP | 557.6 | 560.5 | 2.92 | 561.5 | 3.88 | 561.4 | 3.84 | 560.4 | 2.81 | 8! | 4% |
| IP | 11.9 | 13.1 | 1.14 | 12.9 | 0.97 | 13.1 | 1.18 | 12.5 | 0.58 | 10! | 40% |
| SMSP | 7.2 | 9.0 | 1.79 | 7.7 | 0.57 | 7.7 | 0.54 | 7.3 | 0.17 | 111! | 69% |

**Ablation study**  We conducted an ablation study to show the impact of position embedding and symmetry-aware loss on the evaluation performance. We trained the BASE model using the standard cross-entropy loss with labels $\{X^{(i)}\}$ (without any permutation), and applied multiple techniques to address the drawback (a) using position embedding (PE), and the drawback (b) using either symmetry-aware loss (SAL) or lexicographically ordered label $\bar{X}^{(i)}$ (LEX). Hyperparameters used in the Main results section are adhered and utilized in experiments of this section. We reported the objective value of the best solution produced by different combinations of techniques in Table 2, and we compared the computational results via the average primal gap, see Figure 3. From Table 2, we first notice that by applying only one technique will lead to worse performance comparing against the BASE algorithm. Especially, BASE+SAL produced even infeasible solutions on SMSP dataset, which is denoted by '-' in the table. However, when applying two techniques, the performance is significantly enhanced. By replacing LEX in BASE+PE+LEX by SAL, we achieve the best value among approaches conducted in this section.

It is noteworthy that, from Figure 3, BASE+PE+SAL can retrieve remarkably high-quality solutions on both datasets in very early stages, which indicates that the model trained with SAL+PE can output desirable predictions.

Table 2: Average objective values at 1,000 seconds, comparing different combinations of techniques.

| dataset | BASE | BASE+LEX | BASE+SAL | BASE+PE | BASE+PE+LEX | BASE+PE+SAL |
|---------|------|----------|----------|---------|-------------|-------------|
| IP | 13.22 | 13.18 | 13.32 | 13.30 | 13.10 | **12.50** |
| SMSP | 8.30 | 8.29 | - | 7.75 | 7.71 | **7.34** |

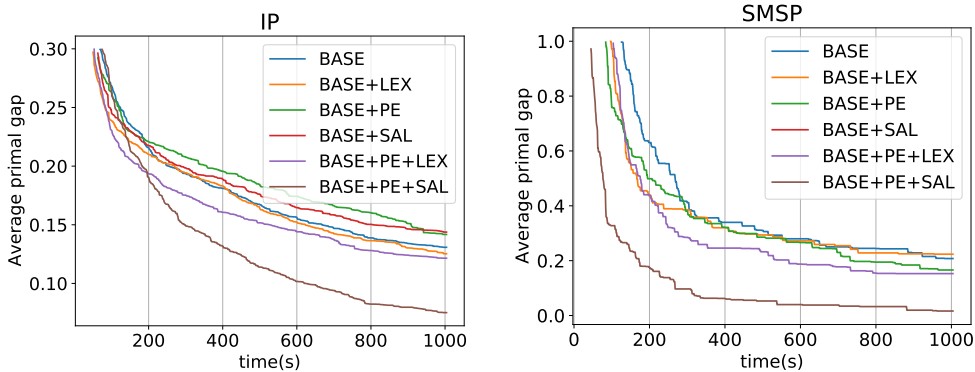

Figure 3: Performance comparison in ablation study.

## 6    CONCLUSION

We proposed a symmetry-aware learning approach for tackling MILPs with symmetry. We first sort out two drawbacks for not handling symmetries in GNNs-based MILP-solving methods. Then, we enable GNNs to differentiate indistinguishable variables by attaching a set of position embedding features to each variable nodes in the original bipartite graph representation. In order to mitigate the ambiguity drawback, we also designed a symmetry-aware loss that updating labels according to GNNs' prediction. By conducting extensive computational studies on public MILP datasets involving symmetries, we demonstrate the effectiveness of our proposed approach. Ablation studies also indicate the individual soundness of two proposed components. Overall, our proposed method achieved significant improvement on primal gaps compared to existing methods.

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

# Supplementary Document
# A Symmetry-Aware Learning Approach for Solving Mixed-Integer Linear Programs

## A  LEXICOGRAPHIC ORDERING

In Section 5.1, the baseline LEX uses the reordered labels $\{\bar{x}^{(i)}\}$ (or their matrix forms $\{\bar{X}^{(i)}\}$) in the training stage. We specify the details of the lexicographical order here. First, let $w, w' \in \{0,1\}^q$, we say $w$ is *lexicographically larger* than $w'$, denoted by $w \succ w'$, if there exists $k \in \{1, 2, \ldots, q\}$, such that $w_i = w_i'$ for all $i < k$, and $w_k > w_k'$. If $w \succ w'$ or $w = w'$, we write $w \succeq w'$. Given the original labels $\{X^{(i)}\}$, the new labels $\{\bar{X}^{(i)}\}$ are generated by reordering the columns of $X^{(i)}$, such that $\bar{X}^{(i)}_{:,1} \succeq \bar{X}^{(i)}_{:,2} \succeq \cdots \succeq \bar{X}^{(i)}_{:,q}$, for all $i$.

## B  HYPER-PARAMETER TUNING IN THE NEIGHBORHOOD SEARCH

The neighborhood search module mentioned in Section 3.4 has three hyper-parameters $k_0$, $k_1$, and $\Delta$ to be tuned, where $\Delta$ restricts the neighborhood radius, $k_0$ and $k_1$ are the cardinalities of index sets $I_0$ and $I_1$, respectively. For simplicity, we use one hyper-parameter $\eta$ (instead of $k_0, k_1$) to control the selection of $I_0$ and $I_1$, which is specified in Algorithm 2. Based on this, we apply the grid search for parameters $\eta \in \{0.1, 0.2, \ldots, 0.9\}$ and $\Delta \in \{0, 5, 10, 15, 20\}$ on a small validation set consisting of eight instances. The hyper-parameters, that produce the smallest average primal gap within 1000 seconds, are selected in different experimental setups and shown in Table 3.

---

**Algorithm 2** Index set selection in neighborhood search

---

1: **Input:** $\hat{x}, \eta$ **Output:** $I_0, I_1$
2: $t \leftarrow 1; I_0 \leftarrow \emptyset; I_1 \leftarrow \emptyset;$
3: **while** $t \leq \eta \cdot s \cdot q$ **do**
4:    $i \leftarrow \arg\max_i |0.5 - \hat{x}_i|,$
5:    **if** $\hat{x}_i < 0.5$ **then**
6:       $I_0 \leftarrow I_0 \cup \{i\};$
7:    **else**
8:       $I_1 \leftarrow I_1 \cup \{i\};$
9:    **end if**
10:   $\hat{x}_i \leftarrow 0.5; t \leftarrow t + 1;$
11: **end while**

---

Table 3: The tuned hyper-parameters $(\eta, \Delta)$ in different experimental setups

|  | AP | IP | SMSP | WA |
|---|---|---|---|---|
| PS | (0.2,15) | (0.1,5) | (0.1,20) | (0.1,15) |
| LEX | (0.1,10) | (0.2,5) | (0.2,15) | - |
| Ours | (0.1,5) | (0.1,0) | (0.5,0) | (0.1,15) |

## C  PREDICTION ERROR

Here we introduce a new measure, the Hamming distance, between a rounded prediction and its nearest symmetric solution. (Symmetric solution is the permuted label according to symmetry.) Specifically, given a prediction $\hat{x}$ and its nearest solution $\tilde{x}$, the top-$k\%$ error is defined as $\mathcal{E}(k) = \sum_{i \in K} |\text{Round}(\hat{x}_i) - \tilde{x}_i|$, where $K$ is the index set of $k\%$ variables with largest values of $|0.5 - \hat{x}_j|, \forall j$. As shown in following Table, our approach always outputs prediction with less error when compared to the PS framework, illustrating that our approach can correctly predict a larger proportion of variables.

| error | IP | | SMSP | | AP | |
|---|---|---|---|---|---|---|
| | PS | Ours | PS | Ours | PS | Ours |
| top-10% | 1.50 | 0.53 | 11.16 | 0.00 | 3.63 | 0.79 |
| top-20% | 3.78 | 1.78 | 23.69 | 0.00 | 8.26 | 4.05 |
| top-30% | 8.57 | 2.88 | 36.05 | 0.00 | 11.00 | 7.47 |
| top-40% | 13.10 | 3.93 | 48.54 | 0.03 | 15.21 | 11.37 |
| top-50% | 21.00 | 5.30 | 61.05 | 1.48 | 18.95 | 14.37 |
| top-60% | 29.52 | 17.37 | 73.44 | 9.10 | 22.00 | 17.53 |
| top-70% | 43.43 | 36.12 | 85.82 | 19.26 | 25.89 | 22.42 |
| top-80% | 59.55 | 56.08 | 98.34 | 31.61 | 28.84 | 26.79 |
| top-90% | 80.22 | 76.30 | 115.62 | 47.33 | 33.58 | 31.21 |
| top-100% | 105.03 | 98.70 | 222.59 | 145.48 | 40.32 | 39.00 |

# D  CASE STUDY

As the GNN can output a value between 0 and 1 for each binary decision variable, we further examine the distributions of GNN predictions produced by different approaches. Hereby, we plot the sorted prediction values of different GNN-based approaches on one IP problem instance, as a case study, see Figure 4. One can observe that BASE approach outputs a very average predictions, providing few information, while both BASE+PE+LEX and BASE+PE+SAL have distinct predictions. This is precisely what we desire, as a good model prediction should be close to 0 or 1. Moreover, we also observed that SAL can provide better predictions compared to LEX, which explains why our model performs the best to some degrees.

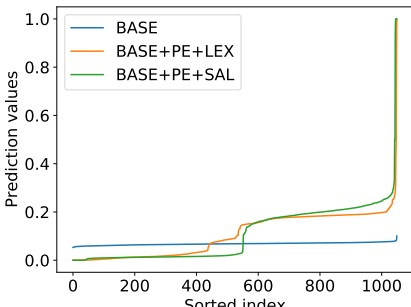

Figure 4: Sorted prediction values produced by different algorithms.

