# OpenReview forum: "A Symmetry-Aware Learning Approach for Solving Mixed-Integer Linear Programs"
_ICLR.cc/2024/Conference — ICLR 2024 Conference Withdrawn Submission_

### Official Review · Reviewer_9XGv · 2023-10-30

**Soundness:** 2 fair
**Presentation:** 3 good
**Contribution:** 2 fair
**Rating:** 5
**Confidence:** 4

**Summary:**

This paper focuses on two issues---indistinguishable variable nodes and label ambiguity---in using GNNs to solve MILPs with symmetries.
It proposes to use position embeddings and designs a symmetry-aware loss to alleviate the aforementioned two issues, respectively.
Experiments on four datasets demonstrate the effectiveness of the proposed method in terms of computational efficiency and solution quality.

**Strengths:**

1.	This paper is easy to follow. It’s worth mentioning that it formulates the issues in using GNNs to solve MILPs with symmetries clearly.
2.	The proposed methods are simple yet effective. They can be combined with a series of GNN based approaches to improve their performance.
3.	Experiments on four benchmark datasets, as well as the extensive ablation studies, demonstrate the effectiveness of position embeddings and the symmetry-aware loss.

**Weaknesses:**

1.	The authors may want to provide more discussions on some related works.

	a) The authors mention that [1] and [2] “try to tackle the label ambiguity drawback by learning conditional distribution” (Section 2). Then, they claim that it “is not symmetry-aware” (Paragraph Baselines, Section 5.1). However, more explanations are necessary.

	b) The authors claim that “the random features used in [2] may cause severe generalization issue” (Section 2). They may want to conduct experiments to support this claim and to demonstrate the superiority of position embeddings over random features.

	c) The authors claim that “none of the existing learning-based approaches take special care of symmetry handling” (Section 2). However, these approaches identified the symmetry issues and proposed some tricks to deal with these issues. The authors may want to further explain what they mean by “take special care of symmetry handling”.

	d) Since “works from mathematical perspectives suggest symmetry-handling algorithms” (Section 2), the authors may want to provide more details about the relationship between this work and those works from mathematical perspectives.

2.	The position embedding technique, which is one of the core contributions of this paper, has been implemented in the code of [3] to tackle the strong symmetry.
https://github.com/sribdcn/Predict-and-Search_MILP_method/blob/main/PredictAndSearch_GRB.py#L47.
Therefore, the technical contribution is minor.
3.	The symmetry-aware loss involves using Hungarian Algorithm to solve Problem (9), whose time complexity is $O(n^3)$. This may lead to a high training cost. Therefore, the authors may want to report the running time.
4.	The authors may want to report the standard deviation of the results in Table 1.
5.	Some of the references, such as [2] and [3], are in the wrong form.

[1] Vinod Nair, Sergey Bartunov, Felix Gimeno, Ingrid Von Glehn, Pawel Lichocki, Ivan Lobov, Brendan O’Donoghue, Nicolas Sonnerat, Christian Tjandraatmadja, Pengming Wang, et al. Solving mixed-integer programs using neural networks. arXiv preprint arXiv:2012.13349, 2020.

[2] Ziang Chen, Jialin Liu, Xinshang Wang, and Wotao Yin. On representing mixed-integer linear programs by graph neural networks. In The Eleventh International Conference on Learning Representations, 2023.

[3] Qingyu Han, Linxin Yang, Qian Chen, Xiang Zhou, Dong Zhang, Akang Wang, Ruoyu Sun, and Xiaodong Luo. A gnn-guided predict-and-search framework for mixed-integer linear programming. In The Eleventh International Conference on Learning Representations, 2023.

**Questions:**

1.	Why are the results of BKS, Gurobi and PS on the datasets WA and IP different from those reported in [3]?
2.	It seems that with the position embedding, the permutation invariance does not hold any more. For example, if we exchange the positions of two different columns, we will obtain different graph embeddings as they will have invariant node features while variant position embeddings. Is that the case?
3.	How to calculate $\Phi$ for a given dataset?
4.	Since $\Phi$ indicates the number of possible equivalent solutions, it measures the extend of the issue of label ambiguity. Is there any metric to measure the extend of the issue of indistinguishable variable nodes?

---

### Official Review · Reviewer_MwS2 · 2023-10-30

**Soundness:** 2 fair
**Presentation:** 2 fair
**Contribution:** 2 fair
**Rating:** 3
**Confidence:** 3

**Summary:**

The paper “ A symmetry -aware Learning Approach for solving Mixed-Integer Linear programs'' discusses  how to solve mixed-integer linear programs (MILPs), while taking into account the symmetry that may exist between variables leading to multiple optimal solutions.  It highlights the issue that MILPs often have symmetry, resulting in multiple equivalent solutions and causing challenges for traditional optimization methods. While machine learning-based approaches using graph neural networks (GNNs) have shown promise in solving MILPs, they haven't addressed the issue of symmetry. To tackle this problem, the paper proposes a "symmetry-aware" learning approach that incorporates position embeddings to distinguish interchangeable variables and introduces a novel loss function to handle the ambiguity caused by equivalent solutions. The authors conducted experiments on public datasets and found that their approach outperforms existing methods in terms of computational efficiency and solution quality.

**Strengths:**

MILP has been used in various capacities in optimization aspects of ML and  recently, with the development of GNN, MILP can be solved as GNNs and this enables use of MILPs as a plug in module in many different deep learning models. In this respect, the paper studies a relevant problem.

**Weaknesses:**

The premise of the paper is based on the weak assumption that symmetry incurs noticeable computational costs. This is not clearly demonstrated in the paper.
The improvements proposed by the paper, are not justified as sound theoretical choices. The position embedding seems to add  additional features(why necessarily this way) whereas the symmetry aware loss minimizes distortion to another permutation of the input. What it means in terms of optimization is not clear maybe a simple example of how this helps will illustrate the point better.
Experimental results are ok, but does not backup the main premise of the the paper that symmetry is significant cost on computational resources.

**Questions:**

NA

---

### Official Review · Reviewer_7CFb · 2023-10-30

**Soundness:** 3 good
**Presentation:** 2 fair
**Contribution:** 1 poor
**Rating:** 3
**Confidence:** 3

**Summary:**

The paper introduces a neural network MIP solver that is designed to handle symmetry in binary linear programming. The solver predicts initial solutions for MILP using neural networks. The authors highlight the significance of handling variable symmetry and suggest addressing this challenge through positional embedding and a custom-designed loss function. Experimental results show that this approach is superior to other neural solvers and the default Gurobi configuration.

**Strengths:**

* The paper is written in a clear and easy to understand manner.
* While the traditional optimization community has widely recognized the symmetry of variables, it has not been studied as much with neural networks. The authors provide valuable insights into the field of neural MILP solving, and their efforts should be greatly appreciated.

**Weaknesses:**

- My primary concern is that this paper represents incremental work compared to a paper published in ICLR 2023 by Han et al., titled "A GNN-Guided Predict-and-Search Framework for Mixed-Integer Linear Programming."
  - The paper by Han et al. is cited and compared as a significant baseline in this paper. Both papers follow a similar process: a neural network predicts an initial solution, certain digits with higher confidence are fixed and treated as new constraints, and finally, Gurobi is used to solve the modified problem with additional constraints. The only technical addition in this paper appears to be the inclusion of positional embeddings in nodes.
  - What makes this paper seem more incremental is my observation that positional embeddings are already implemented in the code accompanying Han et al. (ICLR 2023): [github link](https://github.com/sribdcn/Predict-and-Search_MILP_method/blob/b45ded73d261ef912ebb56934607d6424a88b387/PredictAndSearch_GRB.py#L48). My point is that publishing a new paper at a top ML conference by employing a technique that has already proven effective in a previous paper, without providing valuable insights, is questionable.
  - The experimental improvement compared to Han et al. (ICLR 2023) also seems incremental in Table 1.
- The authors might argue for the importance of symmetry in solving MIP, which I agree with. However, as an ICLR paper, it requires stronger motivation.
  - An excellent example can be found in another paper published at ICLR 2023, where the breaking of symmetry is explored: Chen et al.'s "On Representing Mixed-Integer Linear Programs by Graph Neural Networks." Chen et al. offer theoretical insights into solving symmetric MIPs using graph neural networks. They discovered that symmetry is a theoretical bottleneck of GNNs when solving MIPs, and then proposed to break symmetry by introducing random noise.
  - It's worth noting that when breaking the symmetry of MIPs, positional embeddings and random noise are two sides of the same coin. The underlying technical motivation is to assign the symmetric nodes different features. Also, positional embeddings are random features concatenated to nodes, because positional embeddings are not permutation-invariant. If we randomly permute the nodes, the positional embeddings also change randomly.

**Questions:**

* Seeing that Han et al. (ICLR 2023) have already implemented positional embedding, can the authors explain where the performance improvement comes from?
* Can the authors distinguish this paper, technically, from Han et al. (ICLR 2023)?

---

### Official Review · Reviewer_Vpp5 · 2023-11-03

**Soundness:** 3 good
**Presentation:** 3 good
**Contribution:** 2 fair
**Rating:** 3
**Confidence:** 4

**Summary:**

This paper studies GNNs for solving MILPs and considers the symmetric properties of MILPs (i.e., switching some variables does not essentially change the problem). The proposed methods include: 1) adding additional features (positional embedding) to differentiate variables that are symmetric; 2) defining the loss function for an orbit of the symmetry group, not just a point. The approach is tested on two datasets and compared with other methods. Some numerical advantages are reported by the authors.

**Strengths:**

Symmetry is an important property in MILPs but is rarely considered in GNN-based methods for solving MILPs. It is good to introduce this topic to the community. The writing is clear. The ideas of adding differentiable features and using orbit loss (or symmetry-aware loss) make sense (they are standard in handling symmetry and have been used in other problems). The numerical results show that the approach is promising.

**Weaknesses:**

1. This paper only considers one very special type of symmetry group (a binary matrix and the symmetry group is the the permutation group on the columns). In general, the exact symmetry group of a MILP problem, and hence the symmetry-aware loss, may be very expensive to compute, which makes the proposed method in some sense impractical. I would suggest investigating algorithms based on only partial information about the symmetry group (e.g. the output of the Weisfeiler-Lehman test/color refinement).
2. Line 6 in Algorithm 1 is expensive to implement, especially when the symmetric group is large. I think the authors should discuss how to solve this subproblem (probably approximately) fast, instead of just saying "can be solved in parallel".
3. Positional embedding (or the related random feature technique) has frequently appeared in the previous literature, which makes the contribution of this paper in some sense incremental.

**Questions:**

None.